# Maternal Factors and Their Association with Patterns of Beverage Intake in Mexican Children and Adolescents

**DOI:** 10.3390/children8050385

**Published:** 2021-05-13

**Authors:** Desiree Lopez-Gonzalez, Fatima Avila-Rosano, Diana Montiel-Ojeda, Marcela Ortiz-Obregon, Pamela Reyes-Delpech, Laura Diaz-Escobar, Patricia Clark

**Affiliations:** 1Clinical Epidemiology Research Unit, Hospital Infantil de Mexico Federico Gomez, Mexico City 06720, Mexico; nut.monseavilar@gmail.com (F.A.-R.); nutriologadianamont@gmail.com (D.M.-O.); nutriologamarceortiz@gmail.com (M.O.-O.); pameanadelpech@gmail.com (P.R.-D.); dralmed@hotmail.com (L.D.-E.); 2Faculty of Medicine, Universidad Nacional Autonoma de Mexico, Mexico City 04360, Mexico

**Keywords:** maternal factors, children, adolescents, beverage pattern, added sugar, sweetened beverages

## Abstract

Childhood and adolescence represent critical periods where beverage and food consumption behaviors are learned and developed. Mexican mothers’ presence and influence are instrumental in shaping such behaviors. The aim of this study was to estimate the prevalence and risk associations of maternal factors for unhealthy patterns of beverage intake. This study analyzed data from a population-based cross-sectional study of healthy children and adolescents from Mexico City. Data of subject’s total water intake (TWI) and its’ sources were collected using two 24-h recall surveys. Patterns of beverage intake were constructed based on the guidance system of beverage consumption in the US. Maternal factors of interest included age, body mass index (BMI), mother’s educational level (MEL), socioeconomic status (SES), and belongingness to the paid workforce (BPW). Data of 1532 subject–mother dyads informed that 47% of subjects did not meet the Institute of Medicine (IOM) recommendations for TWI, and 94.6% showed an unhealthy beverage intake pattern, mainly consisting in a lower intake of water and a higher intake of caloric beverages with some nutrients; and calorically sweetened beverages. The major sources of hydration were caloric beverages with some nutrients (i.e., whole milk, fruit water, and flavored milk). The highest risk association for an unhealthy beverage intake pattern was seen in those subjects with mothers in the cluster with lower SES, lower MEL, lower proportion of BPW, higher BMI, and younger age (OR = 9.3, 95% CI 1.2–72.8, *P* = 0.03). Thus, there is a remarkably high prevalence of an unhealthy pattern of beverage intake, and specific maternal factors may be implicated as enablers of such behaviors, which is also addressable for future interventions.

## 1. Introduction

Hydration status is a significant health indicator and directly correlated with healthy metabolism [1,2]. Water is critical for multiple physiological functions, and the human body must maintain constant equilibrium between water intake and water loss [3].

Water intake behaviors vary among individuals depending on age, gender, culture, education level, and availability of water sources, among other factors [4]. Evidence correlates adequate water intake with high–quality diets, better lifestyle habits, and a lower risk of chronic diseases [5]. The consumption of sugar-sweetened beverages has been associated with cardiometabolic risks among adolescents [6].

The United States Institute of Medicine (IOM) has published recommendations for an adequate daily water intake for different sex and age groups [4]. Knowledge and compliance with these recommendations contributes to healthy hydration status. However, IOM recommendations focus on the quantity of total water intake (TWI), not the quality of its sources [4].

The sources of water are fluids and water from food. The fluids are composed of different types of beverages and have been classified according to their nutritional and calorie contents by the “Guidance System for Beverage Consumption in the U.S.” [7]. This classification ranks beverages by their nutrients and calories into six levels. The hierarchy of this classification relates to the quality and nutrient bioavailability from these beverages [8,9,10]. A healthy and balanced pattern of beverage consumption should reflect most intake from plain water, infusions and infrequent fat and skim milk (levels 1–3) and low or non-existent intake from sweetened beverages (levels 4–6). These last levels include beverages with high caloric density, low nutritional content, or both, and they have known associations with negative health outcomes [11]. The guidance system recommends a suggested and an accepted beverage pattern that would provide at most 10% and 14% of total daily energy intake coming from beverages respectively [7]. Risk associations of dietary added sugars have also been addressed by the American Heart Association (AHA) with the specific recommendation of not exceeding 25 g of added sugar intake per day (irrespective of their source) [11].

Childhood and adolescence represent critical periods where eating behaviors are learned and developed [12]. These behaviors have the potential to affect future adult health [13]. As in many other cultures, Latin mothers’ presence and influence are instrumental in shaping eating behaviors. Several maternal factors have been previously associated with specific patterns of beverage intake in children and adolescents [14]. Previous studies in Mexico have documented a high intake of sweetened beverages starting early in life [15]. The relationship between maternal factors and the pattern of beverage intake in Mexican children and adolescents has not been previously reported.

The aim of this study is to estimate the prevalence and risk associations of maternal factors for unhealthy pattern of beverage intake in Mexican children and adolescents.

## 2. Materials and Methods

### 2.1. Study Participants

Data for the current analysis came from the “Reference Values of Body Composition in Pediatric Mexican Population” study, a population-based cross-sectional study of healthy children and adolescents from Mexico City [16,17]. Briefly, this study was designed to construct normative data for body composition for healthy Mexican children and adolescents. Subjects were recruited through a randomized stratified multistage procedure to represent such population. They were clinically and nutritionally assessed to confirm their healthy status and provide data to construct reference values. For this study, subjects that went to the study assessment accompanied by their mothers, the participants that did not have known chronic, endocrine, systemic, respiratory, neurological, cardiac, or psychiatric disorders, nor chromosomal diseases, genopathies, dysmorphic syndromes and that provided informed consent/assent were included. This study was reviewed and approved by our Institutional Research, Ethics and Biosafety Committees (registry no. HIM 2015–055).

### 2.2. Measurements

Clinical measurements of subjects and their mothers included weight and height, measured with subjects wearing lightweight clothing using a SECA^®^ 284 (seca gmbh & co., Hamburg, Germany) scale with stadimeter, measurements were standardized and performed by trained nutritionists. The body mass index (BMI) was calculated as the weight (kg) divided by the square of height (m) [18]. Subjects were further classified according to their BMI percentile value, based on the growth charts of the World Health Organization (WHO), into underweight (< 5th percentile), healthy weight (5th to <85th percentile), overweight (85th to <95th percentile), and obese (≥ 95th percentile) [19]. Mothers were classified according to conventional adults’ cutoff values into healthy weight (BMI 20 to <25 kg/m^2^), underweight (BMI < 20 kg/m^2^), overweight (BMI 25 to <30 kg/m^2^), and obesity (BMI ≥ 30 kg/m^2^).

#### 2.2.1. Beverage, Water, Energy and Added Sugar Intake

Data on quantity and types of food and beverages intake was collected by trained nutritionists through a structured interview to each mother–subject dyad using two 24-h recall surveys, one administered to the nearest past typical school–day and the other to the nearest past typical weekend day. A weighted mean from both surveys was calculated for corresponding analyses. A 12-month food frequency questionnaire (FFQ) was also applied. Briefly the FFQ includes 133 food items in the following categories: dairy products, carbohydrates, fats, proteins, vegetables, fruits, water, beverages with and without added sugar, and highly processed calorie dense foods (i.e., sweetened beverages, candies, fast food, cakes, etc. were adapted for the pediatric population) [20]. This FFQ has been previously validated and used to assess and inform diet habits in the Mexican population [21]. For this study, reported raw data was used to estimate the total water intake (TWI), total daily energy intake (TDEI), and total added-sugar intake from beverages (TASI) by means of the Food Processor Software version 11.1^®^ ( ESHA research, Oak Brook, IL, USA). and for local products we used the published Mexican equivalents [22]. 

Misreporting bias for this sample was performed applying the Goldberg cut–off method adapted for children by Black [23] and previously published by our group. Briefly, 15% of the sample was classified as over-reporting, 3% as under-reporting, and 81% as plausible reports [20].

All reported beverages and their quantities where captured and classified according to the following 6 levels [7]:

Level 1. Plain water: tap and bottled water.

Level 2. Infusions: tea or coffee without sugar.

Level 3. Low-fat & skim milk: skim milk, semi–skim milk, light milk, and low-fat drinkable yogurt without sugar.

Level 4. Non-calorically sweetened beverages: infusions, flavored water, soft drinks, and industrialized drinks sweetened with non-caloric sweetener.

Level 5. Caloric beverages with some nutrients: whole milk, flavored milk, sugary drinkable yogurt, natural juices, industrialized juices, sports drinks, industrialized prepared water, fruit water, and nectars.

Level 6. Calorically sweetened beverages: soda, frappe drinks, energy drinks, infusions with sugar, and jelly.

#### 2.2.2. Patterns of Beverages Intake

The beverage patterns were constructed based on the guidance system of beverage consumption in the US [7]. This system ranks beverage in six levels, from the beverages that should be consumed in limited quantities (Level 6) to the beverages that should be consume as the main source (Level 1, i.e., water). The guidance system recommends a suggested and an acceptable beverage consumption pattern. An extrapolation of the relative proportions for each level was applied to the TWI IOM recommendations for children and adolescents to assess compliance or not to such patterns [4]. An unhealthy beverage intake pattern was defined as not complying to either the suggested or acceptable patterns, mainly consisting in a lower intake of water and higher intake of inadequate sources of water (exceeding the daily upper limit from non-calorically sweetened beverages, caloric beverages with some nutrients, and calorically sweetened beverages).

Acknowledging a differential potential impact of complying or not to the different levels recommendations and incorporating the IOM and AHA recommendations on TWI and TASI respectively, we built a “suggested beverage intake score” (SBIS) and an “acceptable beverage intake score” (ABIS) with different weights for each of the recommendations. The IOM, AHA, level 1, and level 6 recommendations were prioritized and weighted as 20% each, followed by recommendation on level 5 beverages with 10%, and recommendations on level 2–4 beverages with 3.3% each. This approach could result in specific scores for each subject between 0 and 1, where 0 means not complying to any of the 8 recommendations and 1 complying to all of them. Differences between SBIS and ABIS consisted in the different criteria used for suggested or acceptable beverage intake patterns according to the guidance system of beverage consumption in the US [7]. 

#### 2.2.3. Maternal Factors

Data on mothers’ educational level (MEL) was collected and categorized as (a) elementary school (+6 years), (b) secondary school (+3 years), (c) high school (+3 years), (d) a technical degree (+1–3 years), (e) a bachelor’s degree (+4–5 years), and (f) a master’s or doctoral degree (+1–4 years). Socioeconomic status (SES) was determined as the total monthly income per family and categorized according to the classification of the “Asociación Mexicana de Agencias de Inteligencia de Mercado y Opinión (AMAI)” in Mexican pesos [24]. This value was then converted into American dollars and grouped using the following AMAI classification: A/B = upper class (> USD $2225.00); C+ = high middle class: (USD $1978.00–2224.00); C = middle class: (USD $1334.00–2225.00); C− = emerging middle class: (USD $801.00–1333.00); D+ = medium–low class (USD $401.00–800.00); D = low high class (USD $200.00–400.00); and E = lower class: (USD < $200.00) [24]. For the purpose of this study, families were categorized as upper class (i.e., groups A/B and C+), middle class (i.e., C, C−, and D+), and lower class (i.e., D and E). Other relevant variables included mothers’ age, and belongingness to the paid workforce (BPW) defined as being remunerated for their work. 

### 2.3. Statistical Analysis

Conventional descriptive statistics were used to inform regarding the characteristics of the subjects, their mothers and studied variables of TWI, frequency of beverages intake, TDEI, daily caloric intake coming from beverages, TASI and maternal factors. Because TDEI and TWI requirements differ by age and sex, data were disaggregated into three age groups: (a) young children (4–8 years old), (b) children (9–13 years old), (c) and adolescents (14–18 years), and each into males and females. 

The reported data on beverage consumption were used to estimate the total water intake (TWI), which was compared with the IOM age–specific recommendations for adequate intake [4], only for fluids [25,26]. As follows: for children 4–8 y (1200 mL), males 9–13 y (1800 mL), males 14–18 y (2600 mL); females 9–13 y (1600 mL), females 14–18 y (1800 mL). 

Relationships between maternal factors and non–compliance to the different recommendations, and SBIS/ABIS were assessed by univariate and multivariate logistic and linear regression analyses, adjusting for age and sex of the subjects.

Two-step cluster analyses, taking the maternal factors MEL, SES, and BPW as independent categorical multinomial variables, mothers’ age and BMI as standardized continuous variable, and the number of clusters automatically determined based on log-likelihood distances were done. The resulting clusters were used to explore the specific combinations of maternal factors that may discriminate different risk associations to an unhealthy beverage intake pattern, lower SBIS and/or ABIS.

All statistical analyses were performed using IBM^®^ SPSS Statistics, version 20.0 Armonk, NY: IBM Corp. without replacement of missing values. Values of *P*  < 0.05 were considered statistically significant.

## 3. Results

This study included data on 1532 mother–subject dyads. Sociodemographic and anthropometric data are summarized and presented in Table 1 disaggregated by subjects’ age and sex groups. Subject’s data on weight, height, and BMI were comparable to those of ENSANUT 2018 for similar age groups, providing evidence of the representativeness of the sample (comparison not shown) [17,27].

Eighty one percent of the sample consumed beverages from level 1, 3% from level 2, 17% from level 3, 7% from level 4, 94% from level 5, and 31% from level 6. A detailed summary of all types of consumed beverages disaggregated by age– and sex–groups is presented in Appendix A. Detailed data on TWI, its sources’ distribution across the six levels of beverages, TASI, TDEI, and daily caloric intake from beverages are presented in Table 2. 

Considering the total sample, 812 (53%) subjects comply with the IOM recommendations for adequate TWI. More than the half of the sample (829, 54%) exceeded the AHA TASI recommendation. Most of the subjects, a total of 1449 (94.6%) showed an unhealthy pattern according to non–compliance to neither of the suggested nor acceptable recommendations of the US Guidance System. Remarkably, only nine subjects (0.6%) showed a healthy beverage pattern, consisted with the compliance with the 8 recommendations (IOM, AHA, and levels 1–6 guidance system suggested pattern). Details of these results disaggregated by sex and age groups are presented in Table 3.

### 3.1. Beverage Sources

Considering the total sample, major sources of hydration were levels 5 (caloric beverage with some nutrients) and 1 (water) with 46% and 34% of TWI, respectively. Level 6, 2, 4, and 3 beverages represented 7%, 4%, 2%, and 0.5% of TWI, respectively. Data for the different age and sex groups did not differ from the total sample. A graphic representation of this distribution for the total sample and disaggregated by age and sex groups is presented and contrasted to suggested and acceptable patterns in Figure 1. 

Median SBIS for the total sample was 0.5 (IQR 0.3–0.7), and for ABIS was 0.6 (IQR 0.47–0.7). Specific data on each age– and sex–groups are shown in Table 4.

### 3.2. Maternal Factors 

Prevalence and distribution of maternal factors are summarized in Table 1. BMI, BMI category, MEL and BPW showed similar values to those for Mexico City and Metropolitan Area estimated by ENSANUT [27,28], the Mexican National Institute of Geography, Statistics, and Informatics (INEGI) and the Mexican National Employment and Occupation Survey (ENOE) [29]. However, our sample showed lower proportion of members of the high class (5% vs. 19%), higher proportion of the middle class (53% vs. 45%) and higher proportion from the lower class (42% vs. 37%) compared to nation–wide data published by AMAI [24]. Significant positive relationships were found between age, MEL, SES, and BPW. Mothers’ BMI showed a significant negative relationship with MEL and SES, and positive for mothers’ age (Appendix A). 

Univariate logistic regression of maternal factors (age, BMI, MEL, SES, and BPW) for non-compliance to each of the eight recommendations indicated that increasing age of mothers showed a significant risk association for non–compliance to the IOM TWI recommendation (OR = 1.04, 95% CI 1.029 to 1.06; *P* < 0.0001). In contrast, increasing MEL and BPW were associated to a decreased risk association for non–compliance to level–6 recommendation (OR = 0.892, 95% CI 0.826 to 0.964, *P* = 0.004, and OR = 0.747, 95% CI 0.606 to 0.92, *P* = 0.006). BWP also showed a decreased risk association for non–compliance to level–1 recommendation (OR = 0.765, 95% CI 0.615 to 0.952; *P* = 0.017). Finally, increasing maternal BMI showed a decreased risk association with non–compliance to level–5 recommendation (OR = 0.967, 95% CI 0.942 to 0.993; *P* = 0.014). Multiple regression analysis of all maternal factors and adjusted by sex and age of the subjects for each of the 8 studied recommendations, confirmed a significant increased risk association of age of mothers for non–compliance to the IOM TWI recommendation (OR = 1.028, 95% CI 1.001 to 1.057; *P* < 0.045). In contrast, age of mothers showed significant decreased risk associations for non–compliance for level 1 and 6 recommendations (OR = 0.92, 95% CI 0.879 to 0.962; *P* < 0.0001 and OR = 0.964, 95% CI 0.936 to 0.992; *P* = 0.012 respectively).

Multivariate logistic regression for an unhealthy beverage intake pattern as dependent variable and adjusted by age and sex of subjects revealed no independent association of any maternal factor.

Univariate linear regression analyses of maternal factors to SBIS showed that MEL and BWP were significantly associated to higher scores (– coefficients 0.009, 95% CI 0.001 to 0.018; *P* = 0.037 and 0.024, 95% CI 0.001 to 0.047; *P* = 0.039 respectively). BWP kept statistically significant association for ABIS (– coefficient 0.03, 95% CI 0.007 to 0.054; *P* = 0.011) and MEL showed only a tendency (– coefficient 0.008, 95% CI 0 to 0.017; *P* = 0.056). Corresponding multivariate linear regression analyses adjusted for age and sex of the subjects showed no independent associations for any of the maternal factors.

Two-step cluster analyses revealed two different groups according to specific combinations of maternal factors. Cluster 1 was characterized by mothers with lower SES, lower MEL, lower proportion of BPW, higher BMI, and younger age. Cluster 2 was characterized by higher SES, higher MEL, higher proportion of BPW, lower BMI, and older age, as shown in Table 5.

A significantly higher risk association for an unhealthy beverage pattern was observed in those subjects with mothers belonging to cluster 1 (OR = 9.126, 95% CI 1.162 to 71.669; *P* = 0.035). This observation kept statistical significance after adjusting for age and sex of the subjects (OR = 9.259, 95% CI 1.178 to 72.787; *P* = 0.034). Details on each cluster are presented in Table 6. No significant associations were observed between clusters and SBIS nor ABIS.

## 4. Discussion

This study examined the patterns of beverage intake in a representative sample of healthy Mexican children and adolescents from the Metropolitan Area of Mexico City, and estimated the prevalence and risk associations of specific maternal factors for corresponding unhealthy patterns. 

In our study, 1453 (95%) of studied subjects did not comply to either suggested or acceptable recommendations from the Guidance System for Beverage Consumption in the U.S. The three more common recommendations that where not complied where not drinking enough level 1 beverages, and exceeding level 5 and 6 beverages. Seven hundred and twenty subjects (47%) consumed less than the IOM’s daily TWI recommendation and 829 (54%) exceeded the AHA daily TASI recommendation. Our results suggest that the beverage intake patterns in Mexican children and adolescents are in most cases inappropriate and may expose them to future risks of negative health outcomes. 

Other authors have previously reported mean daily TWI for Mexican children and adolescents with variable results. Barquera et al. analyzed nation–wide data from the Mexican National Health and Nutrition Surveys of 1999 and 2006 (ENSANUT) and reported a mean daily TWI of 794 mL for young children and 1254 mL for children, which are lower compared to our observations of 1457 and 1718 mL respectively [30]. Piernas et al. also analyzed nationwide data from ENSANUT 2012 and reported a mean daily TWI of 922 mL per capita in the 4–8–year–old group, 1061 mL per capita for the 9–13 year–old group, and 1390 mL per capita for the 14–18 year–old group, again, compared with our corresponding observations of 1617 mL 1802 mL, and 1922 mL, seemingly lower [31]. Piernas et al. also published specific data for children and adolescents aged 1–18 years old from Mexico City and reported a mean daily TWI and plain water intake of 1094 ml and 411 mL, respectively, which again, compared to our data of 1771 and 743 mL respectively, are lower [31]. We believe that these differences could be explained because of several factors, including the study of different populations (i.e., ENSANUT did not focus on healthy subjects, while we did), the use of different measurement instruments (i.e., ENSANUT initially using a single 24-h recall survey then changing to a semiquantitative food–frequency questionnaire with a limited variety of 17 types of beverages and our study using two 24-h recall surveys without restriction in types of beverages), examinators profiles (i.e., ENSANUT used standardized pollsters, and our study used standardized nutritionists), potential misreporting biases not addressed in ENSANUT data, and finally a potential effect of the different geographical coverages (i.e., ENSANUT being a nationwide study and ours limited to the Metropolitan Area of Mexico City).

More recently a multinational study that included data from Mexico, collected information on beverage intake through a validated liquid intake seven-day record [32]. This study included 669 Mexican children and adolescents and reported a mean daily TWI of 1350 mL for Mexican children aged 4–9 years and 1510 mL for Mexican adolescents aged 10–18 years, which again compared to our results of 1750 and 2200 mL respectively seem lower [32].

Irrespective of which of the previous discussed studies may reflect more precise data, all of them report consistently that Mexican children and adolescents report an average daily TWI below the age and sex–specific IOM recommendations [4]. 

These observations also applied when we compared our data to those previously reported for adolescents by a multinational European study, where daily TWI was of 1611 and 1316 mL for adolescent males and females respectively, compared to our data of 1970 and 1750 mL, respectively [33].

Interestingly, we observed a decreasing rate of compliance with the IOM recommendations for daily TWI in the older age groups, with 82% compliance in young children, and 60% compliance in adolescents. Similar findings were previously reported in an analysis of Mexican children and adult subgroups, wherein 60% of children and 43.3% of Mexican adults were reported as compliant [34]. These findings may reflect differences in hydration behavior due to stronger parental influence on younger children compared with increasing autonomy of adolescents and young adults. 

Regarding the sources of hydration, our main observations are consistent with other data informing that caloric beverages with some nutrients (level 5) are the major source of hydration of Mexican children and adolescents [30,35]. These observations may be contrasted with those of the French population where they have reported that plain water represents their major source of hydration [36]. Such behavior in our sample reflected in the total energy intake coming from beverages representing 22%, 20%, and 19% of the TDEI in young children, children, and adolescents, respectively. These figures are also consistent with those previously published by ENSANUT (28% and 21% for children and adolescents respectively) [30] and significantly above the maximum 14% recommended by the Guidance System for Beverage Consumption in the U.S. [7]. Such behavior also echoed with the 54% of the sample exceeding the AHA daily TASI recommendation, and like our observations of IOM TWI, we saw an increasing effect of age going from 46% to 57% and 61% for young children, children, and adolescents, respectively. Taken together, these observations provide evidence that Mexican children and adolescents are not only, not complying with TWI recommendations, but are hydrating mainly from inadequate sources with resulting increased risk of overweight, obesity, and metabolic and cardiovascular disease [11]. In fact, because this estimation only considers the added sugar from beverages, it may underestimate the proportion of subjects at risk.

Although in the level five include beverages with nutrients, as whole milk and 100% juices, the recommendation is not exceeding the consumption of this beverages because they have a high calories content that could lead to a higher daily caloric intake. According to the American Academy of Pediatrics, the recommendation for children and adolescents is to use non-fat (skim) or low-fat milk, and fruit juice offers no nutritional benefits over whole fruit for infants and children, and has no essential role in healthy, balanced diets.

Univariate and multivariate lineal and logistic regression analyses that looked for potential independent associations between maternal factors and specific behaviors on beverages intake provided marginal associations with a few of them reaching statistical significance. Relevant to such findings McLeod et al. reported that Australian first–time mothers from major socioeconomic position had more knowledge about nutrition in comparison to those mothers with minor socioeconomic level [37]. Campbell et al. analyzed the association between food availability and nutrition knowledge in mothers and they reported that maternal knowledge about nutrition is inversely proportional to the soft drink’s consumption in their children [38]. Tovar et al. also reported that, among other maternal variables, lower maternal age, lower socioeconomic status, and lower education were independently associated to increased risk of early and inappropriate introduction of 100% fruit and vegetables juices and sugar–sweetened–beverages in their infants [39].

In contrast to our findings, several authors, mainly from the United States of America, have reported that maternal fulltime employment may be associated with more unhealthy eating behaviors [40,41]. Different hypotheses have been proposed, including that these mothers may spend less time in meal preparation and recur more to fast foods, school–provided meals, or processed foods than non–working mothers [40]. Moreover, full–time working mothers may be less available to spend time with their children, and these may be exposed significantly more time watching television and the potential negative influence of this [42].

Given the significant relationships between maternal factors, we considered of greater value the identification of specific combinations of these that may discriminate different risk associations to an unhealthy beverage intake pattern. The cluster analysis identified such groups with higher risk towards an unhealthy beverage intake pattern characterized by subjects with mothers of younger age, higher BMI, lower SES, lower MEL, and lower proportion of BPW (OR 9.259, 95% CI 1.178 to 72.787; *P* = 0.034). We believe that the defining characteristics of the two clusters of mothers identified in our sample, remark the relevance of societal disparities and their relationship to significant risk association to unhealthy beverages intake patterns. In our sample the combination seen for cluster 2 may speak of mothers that are less aware of healthier nutritional habits because of their lower education level, being at greater economic difficulties to provide appropriate care because of their lower SES, and less access to regular healthcare provided by social security because of their lower rate of BPW.

Limitations of our study include its cross–sectional nature, recruitment in a single–city with urban setting, and data collection via interview through 24-h recall surveys. The cross-sectional nature limits the scope of the causal relationships between maternal factors, unhealthy behaviors, and ulterior negative health outcomes. However, our results are consistent with those of other authors, and thus provide a clear and consistent picture of a predominant unhealthy pattern of beverage intake in Mexican children and adolescents. Important to note is that this piece of evidence fits with previous evidence of causality between inadequate sugar–sweetened beverages consumption and increased cardiometabolic risk, and also with the overwhelming prevalence of overweight, obesity, and cardiometabolic diseases occurring in our population [43]. We also believe that the characterization of two different risk groups according to maternal factors may provide a setting for potential hypotheses to explore in future studies aiming to improve such interrelated variables that speak of societal disparities. Despite the fact that the sample in this study corresponds only to the central–urban area of the country, represented by Mexico City, previous nationwide published data from ENSANUT did not find significant differences between urban and rural areas in patterns of beverages consumption [44]. Finally, 24-h recall surveys may be susceptible to several biases (e.g., misreporting, memory, response, social desirability, social approval, among others), which we tried our best to avoid by having them applied by standardized nutritionists, included portion size confirmation with standardized materials, collected data spanning > 3 full years, considered week and week-end days for all the samples, and analyzed for misreporting bias. Other instruments, like Liq–In7 and/or the combination of methods, including doubly labeled water, may offer greater precision, but they are also more resource-demanding and in our context, this was not feasible. Other limitation of this study is that we did not measure the water intake from food sources. TWI estimates were based on fluid intake, and IOM recommendations considered were also exclusive for water intake from liquid sources.

## 5. Conclusions

An estimated 95% of children and adolescents from Mexico City reported an unhealthy pattern of beverage intake, mainly driven by a preference of caloric beverages with some nutrients (level 5) as the major source of water intake. Fifty four percent of subjects exceeded the AHA daily TASI recommendation, and 47% did not meet the daily TWI recommended by the IOM. Subjects with mothers of lower SES, lower MEL, lower proportion of BPW, higher BMI, and younger age showed a significant higher risk association for such unhealthy beverage pattern intake. 

There is a remarkably high prevalence of an unhealthy pattern of beverage intake, and specific maternal factors may be implicated as enablers of such behaviors, and also addressable for future interventions

## Figures and Tables

**Figure 1 children-08-00385-f001:**
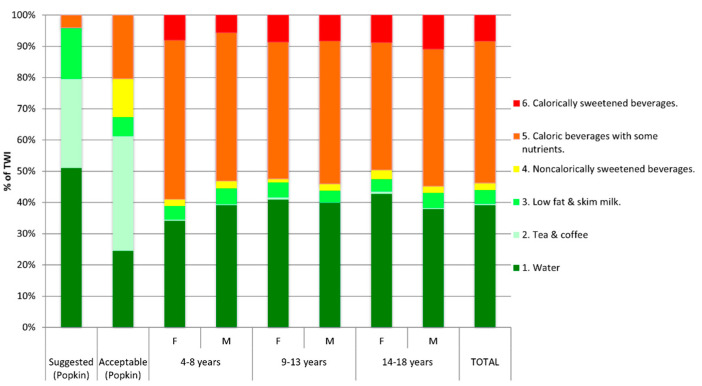
Main sources of TWI un total sample and by age and sex groups. Abbreviations F: female, M: male, TWI: total water intake

**Table 1 children-08-00385-t001:** General characteristics of the sample, categorized by age group and sex.

Variable	Total Sample	Children 4–8 y		Children 9–13 y		Adolescent 14–18 y	
		(*n* = 526; 34%)		(*n* = 588; 38%)		(*n* = 417; 27%)	
	*n* = 1532	Female	Male	Female	Male	Female	Male
		*n* = 230; 53%	*n* = 296; 47%	*n* = 272; 45%	*n* = 316; 56%	*n* = 208; 50%	*n* = 209; 50%
Age (years ± S.D.)	11.1 ± 3.7	6.9 ± 1.2	7.2 ± 1.2	11.5 ± 1.4	11.4 ± 1.5	15.9 ± 1.1	15.8 ± 1.2
Weight (kg ± S.D.)	41.5 ± 17.9	23.9 ± 6.8	25.2 ± 7.4	44.1 ± 13.8	42.9 ± 12.9	57.9 ± 11.1	62.3 ±13.3
Height (cm ± S.D.)	141.6 ± 19.4	118.4 ± 9.4	121.4 ± 9.1	145.7 ±10.0	145.8 ±11.1	156.1 ± 6.3	168.5 ±6.5
BMI (kg/m^2^ ± S.D.)	19.6 ± 4.5	16.8 ± 2.8	16.8 ± 3.1	20.3 ± 4.5	19.8 ± 4.2	23.3 ± 3.9	21.9 ± 4.2
BMI WHO classification							
Underweight (*n*, %)	71 (5%)	12 (5%)	18 (6%)	9 (3%)	20 (6%)	1 (0%)	11 (5%)
Healthy weight	983 (64%)	154 (67%)	201 (68%)	164 (60%)	184 (58%)	136 (65%)	144 (69%)
Overweight	246 (16%)	33 (14%)	36 (12%)	51 (19%)	48 (15%)	52 (25%)	26 (12%)
Obesity	231 (15%)	31 (13%)	41 (14%)	48 (18%)	64 (20%)	19 (9%)	28 (13%)
Puberal Tanner stage							
1 (*n*, %)	725 (47%)	222 (97%)	296 (100%)	49 (18%)	158 (50%)	0	0
2	183 (12%)	8 (3%)	0	80 (29%)	91 (29%)	0	3 (1%)
3	189 (12%)	0	0	91 (33%)	50 (16%)	13 (6%)	35 (17%)
4	285 (19%)	0	0	51 (19%)	16 (5%)	116 (56%)	102 (49%)
5	150 (10%)	0	0	1 (0%)	1 (0%)	79 (38%)	69 (33%)
**Maternal factors**							
Mothers´ age (years ± S.D.)	38.7 ± 7.1	35.2 ± 5.9	35.0 ± 6.6	39.0 ± 6.7	39.0 ± 6.7	42.8 ± 6.7	42.7 ± 5.8
Mothers’ BMI (kg/m^2^ ± S.D.)	27.3 ± 4.6	26.9 ± 4.6	26.7 ± 4.6	27.5 ± 4.7	27.5 ± 4.7	27.6 ± 4.4	27.7 ± 4.5
Mother BMI category							
Under weight	11 (1%)	2 (1%)	5 (2%)	2 (1%)	2 (1%)	0 (0%)	0 (0%)
Healthy weight	484 (33%)	81 (37%)	106 (37%)	86 (33%)	91 (30%)	62 (31%)	58 (29%)
Overweight	619 (42%)	80 (37%)	116 (40%)	110 (43%)	135 (44%)	86 (44%)	90 (44%)
Obesity	356 (24%)	54 (25%)	60 (21%)	61 (23%)	76 (25%)	50 (25%)	55 (27%)
Mother level education							
Elementary school	71 (5%)	11 (5%)	10 (3%)	7 (3%)	18 (6%)	10 (5%)	15 (7%)
Secondary school	334 (22%)	48 (21%)	54 (18%)	55 (20%)	77 (24%)	51 (25%)	49 (23%)
High school	417 (27%)	68 (30%)	85 (29%)	85 (31%)	73 (23%)	50 (24%)	56 (27%)
Post–secondary education	161 (11%)	13 (6%)	25 (8%)	37 (14%)	38 (12%)	28 (13%)	20 (10%)
Bachelor	500 (33%)	79 (34%)	116 (39%)	82 (30%)	96 (30%)	62 (30%)	65 (31%)
Master or equivalent	49 (3%)	11 (5%)	7 (2%)	6 (2%)	14, (4%)	7 (3%)	4 (2%)
Belongingness to the paid workforce	807 (53%)	121 (53%)	153 (52%)	154 (57%)	165 (53%)	107 (52%)	107 (52%)
Socioeconomical level							
Low	310 (42%)	40 (40%)	81 (42%)	57 (44%)	43 (42%)	47 (42%)	42 (42%)
Middle	392 (53%)	53 (53%)	104 (54%)	66 (51%)	53 (52%)	61 (55%)	55 (56%)
Upper	33 (5%)	7 (7%)	8 (4%)	7 (5%)	6 (6%)	3 (3%)	2 (2%)

Values are expressed as means, SD or numbers and percentages. Abbreviations BMI: Body mass index, WHO: World health organization.

**Table 2 children-08-00385-t002:** Total water intake in ml/day, source of water, categorized by age group and sex.

Age Group	Sex	Total Water Intake (TWI) mL		1. Water mL		2. Tea and Coffee mL		3. Low Fat & Skim Milk		4. Noncalorically Sweetened Beverages		5. Caloric Beverages with Some Nutrients		6. Calorically Sweetened Beverages		Total Daily Energy Intake (TDEI) (kcal/d)		Daily Energy Intake from Beverages (kcal/d)		Energy Intake Coming from Beverages (% of TDEI)	Added Sugar Intake from Beverages (g/d)	
		Mean ± S.D.	Median (IQR)	Mean ± S.D.	Median (IQR)	Mean ± S.D.	Median (IQR)	Mean ± S.D.	Median (IQR)	Mean ± S.D.	Median (IQR)	Mean ± S.D.	Median (IQR)	Mean ± S.D.	Median (IQR)	Mean ± S.D.	Median (IQR)	Mean ± S.D.	Median (IQR)	% of TDEI	Mean ± S.D.	Median (IQR)
4–8 y	F	1511 ± 644	1415 (1060–1850)	547 ± 494	500 (250–750)	7 ± 46	0 (0–0)	68 ± 162	0 (0–0)	27 ± 129	0 (0–0)	748 ± 526	628 (450–1000)	113 ± 185	0 (0–250)	2032 ± 533	1969 (1619–2359)	441 ± 195	435 (304–556)	22% (20.7–23.2)	27 ± 25	20 (5–44)
	M	1699 ± 680	1560 (1250–2000)	683 ± 547	560 (250–1000)	4 ± 32	0 (0–0)	77 ± 169	0 (0–0)	42 ± 154	0 (0–0)	794 ± 545	750 (430–1060)	97 ± 188	0 (0–120)	2366 ± 1920	2176 (1749–2597)	475 ± 201	443 (327–600)	22.1% (21.0–23.3)	29 ± 27	22 (9–40)
9–13 y	F	1768 ± 664	1693 (1305–2100)	791 ± 696	628 (250–1100)	10 ± 53	0 (0–0)	72 ± 165	0 (0–0)	21 ± 101	0 (0–0)	736 ± 532	655 (365–1035)	136 ± 228	0 (0–250)	2409 ± 986	2266 (1868–2762)	443 ± 208	440 (300–580)	19.5% (18.4–20.6)	33 ± 32	26 (7–51)
	M	1831 ± 713	1750 (1355–2200)	783 ± 652	750 (250–1175)	4 ± 68	0 (0–0)	63 ± 169	0 (0–0)	43 ± 215	0 (0–0)	790 ± 562	750 (500–1060)	145 ± 276	0 (0–250)	2691 ± 855	2564 (2118–3155)	509 ± 248	470 (350–630)	19.7% (18.7–20.8)	39 ± 36	31 (10–54)
14–18 y	F	1812 ± 650	1750 (1375–2215)	830 ± 681	750 (275–1250)	13 ± 69	0 (0–0)	71 ± 170	0 (0–0)	47 ± 196	0 (0–0)	714 ± 207	610 (318–1015)	136 ± 226	0 (0–250)	2421 ± 1114	2285 (1838–2744)	445 ± 251	431 (252–596)	19.6% (18.2–20.9)	36 ± 35	28 (10–54)
	M	2032 ± 650	1970 (1500–2420)	834 ± 730	750 (250 -1250)	7 ± 46	0 (0–0)	73 ± 179	0 (0–0)	43 ± 175	0 (0–0)	860 ± 606	750 (500–1100)	214 ± 331	0 (0–290)	3579 ± 5656	3017 (2384–3619)	559 ± 299	524 (360–710)	18.3% (17.0–19.5)	48 ± 43	38 (15–73)
TOTAL		1771 ± 774	1650 (1260–2100)	743 ± 642	600 (250–1000)	7 ± 54	0 (0–0)	71 ± 168	0 (0–0)	37 ± 167	0 (0–0)	774 ± 548	735 (423–1050)	138 ± 244	0 (0–250)	2563 ± 2405	2327 (1875–2871)	479 ± 237	450 (308–606)	20.2% (19.7–20.7)	35 ± 33	26 (10–52)

Values are expressed as means, S.D., median and interquartile range (IQR). TWI and beverages levels (1 to 6) are expressed in mL/day. Abreviations: F: female, M: male, TWI: total water intake.

**Table 3 children-08-00385-t003:** Frequency of non–compliance to each of the 8 recommendations disaggregated by age and sex groups.

Age Group	Sex	IOM TWI Recommendation	AHA Sugar Intake < 25g/d	Suggested Beverage Pattern—*n* (%) of Non-Compliant Subjects							Acceptable Beverage Pattern—*n* (%) of Non-Compliant Subjects						
				Full Pattern	Level 1	Level 2	Level 3	Level 4	Level 5	Level 6	Full pattern	Level 1	Level 2	Level 3	Level 4	Level 5	Level 6
4–8 y	F	78 (34%)	101 (44%)	229 (99.6%)	173 (75%)	1 (0.4%)	29 (13%)	13 (6%)	213 (93%)	90 (39%)	227 (99%)	83 (36%)	0	41 (18%)	13 (6%)	199 (87%)	90 (39%)
	M	68 (23%)	141 (48%)	294 (99%)	203 (68%)	2 (0.7%)	42 (14%)	27 (9%)	277 (93%)	98 (33%)	288 (97%)	89 (30%)	0	63 (21%)	25 (8%)	244 (82%)	98 (33%)
9–13 y	F	122 (45%)	144 (53%)	269 (99%)	170 (63%)	2 (0.7%)	36 (13%)	13 (5%)	243 (89%)	103 (38%)	249 (92%)	79 (29%)	1 (0.4%)	51 (19%)	10 (4%)	202 (74%)	103 (38%)
	M	173 (55%)	189 (60%)	311 (98%)	205 (65%)	1 (0.3%)	35 (11%)	21 (7%)	294 (93%)	113 (36%)	295 (93%)	91 (29%)	1 (0.3%)	47 (15%)	21 (7%)	248 (79%)	113 (36%)
14–18 y	F	113 (54%)	119 (57%)	204 (98%)	121 (58%)	3 (1%)	26 (13%)	14 (7%)	185 (89%)	78 (37%)	195 (94%)	56 (27%)	0	37 (18%)	13 (6%)	154 (74%)	78 (38%)
	M	166 (79%)	135 (65%)	206 (99%)	132 (63%)	0	25 (12%)	14 (7%)	192 (92%)	92 (44%)	199 (95%)	74 (35%)	0	35 (17%)	13 (6%)	161 (77%)	92 (44%)
TOTAL		720 (47%)	829 (54%)	1513 (99%)	1004 (66%)	9 (0.6%)	193 (13%)	102 (7%)	1404 (92%)	574 (37%)	1453 (95%)	472 (31%)	2 (0.1%)	274 (18%)	95 (6%)	1208 (79%)	574 (37%)

Values are expressed as numbers and percentages of subjects that no comply with the recommendations. Abbreviations F: female, M: male, IOM: Institute of Medicine, TWI: total water intake, AHA: American Heart Association.

**Table 4 children-08-00385-t004:** Beverage intake score (BIS) by age group and sex.

Age Group	Sex	SBIS		ABIS	
		Mean ± S.D.	Median (IQR)	Mean ± S.D.	Median (IQR)
4–8 y	Female	0.52 ± 0.21	0.5 (0.3–0.7)	0.6 ± 0.22	0.67 (0.5–0.7)
	Male	0.55 ± 0.21	0.5 (0.47–0.7)	0.64 ± 0.21	0.70 (0.50–0.80)
9–13 y	Female	0.51 ± 0.23	0.5 (0.3–0.7)	0.59 ± 0.23	0.6 (0.5–0.77)
	Male	0.47 ± 0.23	0.5 (0.3–0.7)	0.55 ± 0.23	0.53 (0.37–0.7)
14–18 y	Female	0.49 ± 0.23	0.5 (0.3–0.7)	0.57 ± 0.23	0.57 (0.47–0.7)
	Male	0.40 ± 0.21	0.4 (0.27–0.5)	0.47 ± 0.22	0.50 (0.3–0.7)
Total sample		0.49 ± 0.23	0.5 (0.3–0.7)	0.57 ± 0.23	0.60 (0.47–0.7)

Values are expressed as means, S.D., median and interquartile range (IQR). Abbreviations: SBIS = suggested beverage intake score, ABIS = acceptable beverage intake score.

**Table 5 children-08-00385-t005:** Clusters of maternal factors.

	Cluster		
Maternal Factor	Cluster 1 *n* = 335 (47%)	Cluster 2 *n* = 376 (53%)	Factor Weight
SES (*n*, %)	Low (300, 89.6%)	Middle (376, 100%)	1
Average education level (*n*, %)	Secondary (106, 31.6%)	Bachelor (211, 56.1%)	0.23
BPW (*n*, %)	Yes (127, 37.9%)	Yes (243, 64.6%)	0.08
BMI (kg/m^2^) mean ± S.D.	27.5 ± 4.71	26.5 ± 4.34	0.02
Age (years) mean ± S.D.	38.38 ± 7.5	39.1 ± 6.38	0.01

Values are expressed as means, S.D. or numbers and percentages. Abbreviations: SES: socioeconomic status, BPW: belongingness to the paid workforce, BMI: body mass index.

**Table 6 children-08-00385-t006:** Maternal cluster predictor for belong to unhealthy beverage intake.

	OR	95% CI		*p*-Value
*Model 1*				
Maternal Cluster 2 (reference group)				
Maternal Cluster 1	9.126	1.162	71.669	0.035
*Model 2 **				
Maternal Cluster 2 (reference group)				
Maternal Cluster 1	9.259	1.178	72.787	0.034

Risk association, comparison group: mother’s cluster 1. OR = odds ratio, CI = confidence interval. * adjusting for age and sex of the subjects

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
