# Peer review of "Maternal Factors and Their Association with Patterns of Beverage Intake in Mexican Children and Adolescents"

_children, 2021, doi:10.3390/children8050385_

Round 1

Reviewer 1 Report

This is an interesting study looking at patterns of beverage intake in healthy participants and associations of maternal factors that may be related to unhealthy beverage consumption. The article has a strong study design and the findings are significant in terms of both health implications and future interventions in identifying and addressing unhealthy beverage consumption in children.

A couple of suggestions: 

1) Table 3 is a little hard to read and can be formatted to improve legibility 

2) There are multiple minor grammatical errors throughout the paper, I have listed some below, but please proof-read in entirety.  

Line 25: beverage (singular)

Line 44: "Composed of" versus "compound by"

Lines 79-82

130-131 suggested edit: "the system has 6 levels as described above and provides recommendations on the number of ounces per day for each level that an average male..."

Line 205: "did not meet..." Also for this paragraph, "comply with" versus "comply to". 

Line 298: "lower compared to" versus "lower to"

Line 411: adolescents (plural)

3) The authors mention in the discussion that level 5 beverages are the major source of hydration in Mexican children and adolescents. Why may this category of beverage be more popular in this specific population? 

Author Response

Reviewer #1:

This is an interesting study looking at patterns of beverage intake in healthy participants and associations of maternal factors that may be related to unhealthy beverage consumption. The article has a strong study design and the findings are significant in terms of both health implications and future interventions in identifying and addressing unhealthy beverage consumption in children.

A couple of suggestions: 

1) Table 3 is a little hard to read and can be formatted to improve legibility

We have read them and changed or clarified accordingly (in the manuscript changes appear in red).

2) There are multiple minor grammatical errors throughout the paper, I have listed some below, but please proof-read in entirety.  We have corrected as recommended.

Line 25: beverage (singular) Done (line 26)

Line 44: "Composed of" versus "compound by" Done (line 54)

Lines 79-82 We check these lines.

130-131 suggested edit: "the system has 6 levels as described above and provides recommendations on the number of ounces per day for each level that an average male..."

Changed for: The Beverage Guidance System ranks beverage in six levels, from the beverages that should be consumed in limited quantities (Level 6) to the beverages that should be consume as the main source (Level 1, i.e. water)

Line 205: "did not meet..." Also for this paragraph, "comply with" versus "comply to". Done (line 247)

Line 298: "lower compared to" versus "lower to" Done (line 351)

Line 411: adolescents (plural) Done (line 475)

3) The authors mention in the discussion that level 5 beverages are the major source of hydration in Mexican children and adolescents. Why may this category of beverage be more popular in this specific population? 

This could be because in Mexico children and adolescents regularly intake whole milk and regularly add chocolate or sugar to milk, as well as has a regular consume of "fresh water" consisting of water with natural fruit and added sugar, as can be seen in detail in supplementary table 1.

Reviewer 2 Report

Title:  Line 2 beverages should be beverage

Line 10,17:  beverages should be beverage

Line 21:  need to define "unhealthy beverage pattern" before giving statistic

Line 22: source should be sources AND need to state what these beverages are e.g. milk, juice, juice drinks, energy drinks, fortified water etc.

Line 23:  Again need to define unhealthy beverage pattern before stating the association

Line 22-24:  Need p values and r values for associations which were most associated etc.

Line 24:  Still need definition of "unhealthy beverage pattern"  (too much sugar, insufficient milk, other parameters – this is central to the findings so definition is necessary in the abstract

Line 39:  Since this study is conducted in Mexico " The Institute of Medicine (IOM) has published…"  should be "The United States Institute of Medicine (IOM) has published…"

Line 44: "in" should be "from"

Line 44:  the word "compound" does not make sense here "confounded by"? or "composed of"?  -- the meaning is unclear

Line 46:  "bioavailability of" should be "bioavailability from"

Line 50: "low" should be "infrequent"

Line 65-67: Suggest changing " To the best of our knowledge no specific relationships between maternal factors and patterns of beverages intake have been previously reported for the Mexican population."  To "The relationship between maternal factors and the pattern of beverage intake in Mexican children and adolescents has not been previously reported."

Line 69: change " pattern of beverage intake" to " pattern of beverage intake in Mexican children and adolescents."

Line 74: "such study" should be "this study"

Line 77: "such" should be "this"

Line 79-80:  it is not clear whether it is the mothers or the children who do not have any of the diseases or conditions listed – this needs to be clarified

Line 88 and 97:  "standardized nutritionists"  change to "trained nutritionists"

Line 102: change "applied" to "administered"

Line 117-127 – while these categories of beverages have been used others exist as well.  Especially for level 5-6 – many classify unsweetened (no sugar added) juices and 100% juices (rare in the Mexican market) differently from sports drinks flavored milks and other beverages with added sugars.  The reasoning for not separately classifying no sugar added juices needs to be stated in the discussion

Line 77 states that subjects were "clinically, nutritionally, and metabolically" measured yet no metabolic measures are listed in the methods section – either the metabolic measures were inadvertently omitted OR "metabolically" should be deleted from line 77 if they were not performed – note BMI is not a metabolic measure.

Tables 2 and 3 are very difficult  to read (some of the values go over 2 lines)  the word Mean – is divided into Mea then the next line has "n"  the numeric values are similarly nearly impossible to determine.  The tables should be reformatted – possibly separated into additional tables if necessary  but they can not be read in their current format.

Line 282:  beverages should be beverage.

The term unhealthy beverage pattern needs to be precisely defined – it is used throughout the paper and the only definition given is on line 166: "An unhealthy beverage intake pattern was defined 136 as not complying to neither of the suggested or acceptable patterns"  But how much deviation was needed – one extra beverage one time?  Or is it 2 or more times a week from category 6 + 1 time per week or category 5 – or 1 time per month.  Given that this concept is core to the conclusions of the paper stating "unhealthy" without an explicit definition is imprecise and not descriptive of the beverage patterns.

Author Response

Reviewer #2:

Title:  Line 2 beverages should be beverage. Done (line 2)

Line 10,17:  beverages should be beverage. Done (lines 10 and 16)

Line 21:  need to define "unhealthy beverage pattern" before giving statistic. Added in line 21-22: The unhealthy pattern consists in a low intake of water and higher intake of caloric beverages with some nutrients; and calorically sweetened beverages.

Line 22: source should be sources AND need to state what these beverages are e.g. milk, juice, juice drinks, energy drinks, fortified water etc. Done and added (line 22 and 23)

Line 23:  Again need to define unhealthy beverage pattern before stating the association. Done previously (line 21)

Line 22-24:  Need p values and r values for associations which were most associated etc. Added (line 25)

Line 24:  Still need definition of "unhealthy beverage pattern"  (too much sugar, insufficient milk, other parameters – this is central to the findings so definition is necessary in the abstract Done previously (line 21)

Line 39:  Since this study is conducted in Mexico " The Institute of Medicine (IOM) has published…"  should be "The United States Institute of Medicine (IOM) has published…" Added (line 41)

Line 44: "in" should be "from" Changed (line 54)

Line 44:  the word "compound" does not make sense here "confounded by"? or "composed of"?  -- the meaning is unclear Changed (line 54)

Line 46:  "bioavailability of" should be "bioavailability from" Changed (line 58)

Line 50: "low" should be "infrequent" Changed (line 60)

Line 65-67: Suggest changing " To the best of our knowledge no specific relationships between maternal factors and patterns of beverages intake have been previously reported for the Mexican population."  To "The relationship between maternal factors and the pattern of beverage intake in Mexican children and adolescents has not been previously reported." Changed (line 75-76)

Line 69: change " pattern of beverage intake" to " pattern of beverage intake in Mexican children and adolescents." Changed (line 78)

Line 74: "such study" should be "this study" Changed (line 83)

Line 77: "such" should be "this" Changed (line 87)

Line 79-80:  it is not clear whether it is the mothers or the children who do not have any of the diseases or conditions listed – this needs to be clarified. Changed (line 88)

Line 88 and 97:  "standardized nutritionists"  change to "trained nutritionists" Changed (line 97)

Line 102: change "applied" to "administered" Changed (line 125)

Line 117-127 – while these categories of beverages have been used others exist as well.  Especially for level 5-6 – many classify unsweeten.ed (no sugar added) juices and 100% juices (rare in the Mexican market) differently from sports drinks flavored milks and other beverages with added sugars.  The reasoning for not separately classifying no sugar added juices needs to be stated in the discussion  Added (lines 407-413)

Although in the level five include beverages with nutrients, as whole milk and 100% juices, the recommendation is not exceeding the consumption of this beverages, because they have a high calories content that could conduct to a higher consumption of daily caloric intake. According to the American Academy of Pediatrics the recommendation for children and adolescents is to use nonfat (skim) or low-fat milk, and fruit juice offers no nutritional benefits over whole fruit for infants and children and has no essential role in healthy, balanced diets.

Line 77 states that subjects were "clinically, nutritionally, and metabolically" measured yet no metabolic measures are listed in the methods section – either the metabolic measures were inadvertently omitted OR "metabolically" should be deleted from line 77 if they were not performed – note BMI is not a metabolic measure. Eliminated

Tables 2 and 3 are very difficult  to read (some of the values go over 2 lines)  the word Mean – is divided into Mea then the next line has "n"  the numeric values are similarly nearly impossible to determine.  The tables should be reformatted – possibly separated into additional tables if necessary  but they can not be read in their current format. Changed

Line 282:  beverages should be beverage. Changed (line 332)

The term unhealthy beverage pattern needs to be precisely defined – it is used throughout the paper and the only definition given is on line 166: "An unhealthy beverage intake pattern was defined 136 as not complying to neither of the suggested or acceptable patterns"  But how much deviation was needed – one extra beverage one time?  Or is it 2 or more times a week from category 6 + 1 time per week or category 5 – or 1 time per month.  Given that this concept is core to the conclusions of the paper stating "unhealthy" without an explicit definition is imprecise and not descriptive of the beverage patterns.

Mainly consist in a lower intake of water, and higher intake of inadequate sources of water (exceed the daily upper limit from Non-calorically sweetened beverages, Caloric beverages with some nutrients and Calorically sweetened beverages).

Reviewer 3 Report

The authors Lopez-Gonzalez et al. report on the amount and origin of water intake in children living in Mexico-City in relation to characteristics of their mothers. This research is highly relevant as it shows the low adherence to recommended water intake in children and adolescence as well as showing the inadequate sources of water. Most of the water consumed by the study population stems from drinks with nutrients, which are associated with the development of overweight/obesity and the comorbidities. The results further evidence that socioeconomic status and education levels of mothers are major determining factors for children’s eating behavior.

I enjoyed reading the manuscript and applaud the authors for this well written, interesting and relevant manuscript. In the following a few minor comments.

Line 104 – You use the term junk food. As this is not a fixed term and might be interpreted differently I would suggest to either define (e.g highly processed calorie dense foods) or give examples.

Line 205-210 – There is a list of numbers and (%). I am not sure I understand what these numbers refer to. Please rephrase the sentence to make it clearer what these numbers represent. However, as these numbers are given in Table 3 consider interpreting them rather than reciting.

Table 3 is hard to read. Consider tilting the page to increase the readability of the row titles.

Figure 1 is excellent!

I might have missed it but I was missing is a measure of water from food sources. If this was not assessed consider to comment on it in the limitation section.

Minor comments:

Line 88: “these measurements were performed by standardized nutritionists.” I would recommend to write “measurements were standardized and performed by trained nutritionists”

Same in line 97/98 and 405 – I assume you did not standardize the nutritionists.

Line 282 patterns of beverage (minus the s)

Line 286 “did not comply to either” (not neither)

et al. should be written with a full stop

Author Response

Reviewer #3:

The authors Lopez-Gonzalez et al. report on the amount and origin of water intake in children living in Mexico-City in relation to characteristics of their mothers. This research is highly relevant as it shows the low adherence to recommended water intake in children and adolescence as well as showing the inadequate sources of water. Most of the water consumed by the study population stems from drinks with nutrients, which are associated with the development of overweight/obesity and the comorbidities. The results further evidence that socioeconomic status and education levels of mothers are major determining factors for children’s eating behavior.

I enjoyed reading the manuscript and applaud the authors for this well written, interesting and relevant manuscript. In the following a few minor comments. Thank you

Line 104 – You use the term junk food. As this is not a fixed term and might be interpreted differently I would suggest to either define (e.g highly processed calorie dense foods) or give examples. Changed (lines 131-132)

Line 205-210 – There is a list of numbers and (%). I am not sure I understand what these numbers refer to. Please rephrase the sentence to make it clearer what these numbers represent. However, as these numbers are given in Table 3 consider interpreting them rather than reciting. Changed (line 249-251)

Table 3 is hard to read. Consider tilting the page to increase the readability of the row titles. Changed

Figure 1 is excellent! Thank you

I might have missed it but I was missing is a measure of water from food sources. If this was not assessed consider to comment on it in the limitation section. Added (line 469-472)

Other limitation of this study is that we did not measure the water intake from food sources, TWI estimates were based on fluid intake, and IOM recommendations considered were also exclusive for water intake from liquid sources.

Minor comments:

Line 88: “these measurements were performed by standardized nutritionists.” I would recommend to write “measurements were standardized and performed by trained nutritionists” Added (line 97)

Same in line 97/98 and 405 – I assume you did not standardize the nutritionists. Added (line 97)

Line 282 patterns of beverage (minus the s) Changed (line 373)

Line 286 “did not comply to either” (not neither) Changed (line 385)

et al. should be written with a full stop Added
